# Protective role of brain derived neurotrophic factor (BDNF) in obstructive sleep apnea syndrome (OSAS) patients

**Krisstopher Richard Flores**[1]**, Fausta Viccaro**[1]**, Mauro Aquilini**[1]**, Stefania Scarpino**[1]**, Francesco Ronchetti**[1]**, Rita Mancini**[1]**, Arianna Di Napoli**[1]**, Davide Scozzi**[2]**, Alberto Ricci**[1] *

**1** Department of Clinical and Molecular Medicine, Division of Respiratory Diseases, Sant'Andrea Hospital, Sapienza University, Rome, Italy, **2** Department of Surgery, Washington University School of Medicine, St. Louis, Missouri, United States of America

* alberto.ricci@uniroma1.it

**Data Availability Statement:** All relevant data are within the manuscript and its Supporting Information files.

## Abstract

Obstructive sleep apnea syndrome (OSAS) is a common disorder characterized by repeated episodes of upper airways collapse during the sleep. The following intermittent hypoxia triggers a state of chronic inflammation, which also interests the nervous system leading to neuronal damage and increased risk of cognitive impairment. Brain derived neurotrophic factor (BDNF) is a growth factor often associated with neuroplasticity and neuroprotection whose levels increase in several condition associated with neuronal damage. However, whether patients affected by OSAS have altered BDNF levels and whether such alteration may be reflective of their cognitive impairment is still controversial. Here we show that, when compared to healthy control volunteers, OSAS patients have increased serum levels of BDNF. Moreover, OSAS patients with the higher levels of BDNF also have reduced neurocognitive impairment as measured by The Montreal Cognitive Assessment (MoCA) questionnaire. Treatment with standard non-invasive mechanical ventilation (CPAP) also was able to ameliorate the level of cognitive impairment. Altogether our results indicate that BDNF levels represent a neuroprotective response to intermittent hypoxia in OSAS patients.

## Introduction

Obstructive sleep apnea syndrome (OSAS) is a common sleep disorder characterized by repeated episodes of partial or total collapse of the upper airways which result in chronic intermittent hypoxia (CIH), sleep fragmentation, hypercapnia and sympathetic hyperactivity [1]. Neurocognitive impairment is one of the major complications associated with OSAS which also represents a common comorbidity for patients affected by neurodegenerative disease such as Alzheimer and Parkinson [2, 3]. Although sleep fragmentation may partially contribute to cognitive dysfunction, the majority of experimental and clinical evidences indicate that CIH is mainly responsible for the functional and structural brain alterations beyond cognitive

**Funding:** The authors received no specific funding for this work.

**Competing interests:** The authors have declared that no competing interests exist.

dysfunction. In particular, a decreased neuronal excitability in mice hippocampal neurons [4], in addition to increased apoptosis in the cortex [5] and in the CA1 region of the hippocampus [6–8], have all been described in different animal models exposed to cyclic hypoxia. Finally, the increased amyloid-beta (Aß42) found in the brain cortex of mice exposed to CIH, also corroborates the hypothesis that brain alteration induced by OSA would be able to aggravate neural degeneration [9].

At a cellular level, multiple mechanisms such as chronic excitotoxicity from glutamate release [10] and altered vascular relaxation due NO reduction [11], have all been investigated in order to explain the origin of those brain alterations responsible for cognitive dysfunction. However, the most evident explanation is currently represented by the increase in oxidative stress. Several evidences demonstrated that CIH is able to increase Reactive Oxygen Species (ROS) [12] which in turn upregulate a variety of transcription factors such as c-Fos, C-Jun, NFkβ, HIF-1α nd Nrf2 contribuiting to the oxidative stress response [13]. The final imbalance of oxidation-antioxidation lead to a state of oxidative stress, activation of leukocytes, platelets, and endothelial cells with the release of inflammatory cytokines responsible for increased inflammation and cellular apoptosis and necrosis [14, 15].

However, the systemic and neuronal inflammation triggered by the CIH also stimulates those adaptive neuroprotective responses which are required to contain and finally resolve the injury. Brain-derived neurotrophic factor (BDNF) is an essential neurotrophin to neuronal development, survival and differentiation in childhood and adulthood [16–18]. The synthesis of BDNF participates in the recovery from the hypoxic injury by controlling neural plasticity and preventing and/or reducing neuronal death [19]. On the other hand, the lack of BDNF in CIH not only contributes to impaired long-term synaptic plasticity but also fails to prevent neuronal injury, including apoptosis, induced by ROS [20].

In the OSAS, the state of CIH and sleep fragmentation responsible for both systemic and neuronal inflammation contributes to the neurocognitive dysfunction. Considering its role in neuroplasticity after hypoxic injury, BDNF may represent a protective factor released in response to the CIH associated with OSAS. Higher levels of BDNF may be able to slow down the development of cognitive dysfunction observed in OSAS patients by reducing the morphological and functional alterations associated with CIH.

The aim of the present study was to firstly compare the serum BDNF levels between OSAS patients and healthy control volunteers. In addition, we also evaluate a possible correlation between BDNF serum levels and neurocognitive dysfunction. Finally, using non-invasive mechanical ventilation in a continuous positive airway pressure (CPAP) treatment mode we assessed how BDNF serum levels and cognitive dysfunction relate over the time of treatment.

## Materials and methods

### Patient and inclusion criteria

Patients with newly diagnosis of OSAS (n = 24) confirmed after nocturnal cardio-respiratory monitoring, showing an apnea/hypopnea index (AHI)>30 (severe OSAS) and an oxidative desaturation index (ODI)> 30 were enrolled by the respiratory unit day hospital of Sant' Andrea hospital, Sapienza University of Rome. As naïve, OSAS patients had no previous CPAP treatment (T0 group). The control group (n = 10) comprised individuals who were proved to be healthy by continuous nocturnal pulse oximetry and medical examination. The present study was approved by the Institutional Research Ethics Committee of the Istituti Fisioterapici Ospitalieri (IFO) with the approval number 1032/2017. Informed consent was signed from each participant.

All patients were interviewed to collect data concerning sex, age, education, medical history and to record anthropometric characteristics: weight, height, body mass index (BMI), neck circumference and abdomen circumference. Each one underwent global spirometry and arterial blood gas analysis.

The exclusion criteria were current or previous history of cancer, neurodegenerative disturbances, connective tissue diseases, asthma, chronic obstructive pulmonary disease (COPD), chronic hepatitis or gastrointestinal diseases, renal failure, acute or chronic infectious disease, respiratory failure. OSAS patients enrolled received CPAP treatment (T1 group).

## Phase T0 (before-treatment)

In early morning, peripheral venous samplings from naïve OSAS group T0 and controls were collected in serum tubes. The samples were immediately centrifuged at 3000 rpm at room temperature for 13 minutes; supernatants were then collected and stored in aliquots at -80°C until use. All the patients answered to the Montreal Cognitive Assessment (MoCA) questionnaire, a one-page questionnaire designed as a tool for rapid screening of mild cognitive impairment. Different cognitive domains were evaluated: attention and concentration, executive functions, memory, language, constructive visual abilities, abstraction, calculation and orientation. One more point was added to the total score if the patient had 12 years or more of education.

## Interphase

The OSAS group started the recommended therapeutic protocol. Briefly, titration with AUTO-CPAP method for the duration of 3 consecutive nights was performed to reach the therapeutic positive pressures. After appropriate training, a non-invasive ventilation in CPAP-mode was started as home therapy. The patients were weekly contacted to ensure adequate compliance.

## Phase T1 (after-treatment)

After 90 days of CPAP treatment, a second serum sample was collected for all patients and stored at -80 ˚C. On the same day the MoCA test was re-run following the same procedure. In addition, therapeutic compliance (hours of use per night, period of use, air leaks) was assessed checking the data recorded by the ventilator memory card.

## Serum BDNF asssay

Serum BDNF was tested using enzyme linked immuno-absorbent assay (ELISA) according to the manufacturer's instructions (BDNF Emax Immunoassay System; Promega, Madison, WI, USA). Briefly, a specific two-site ELISA assay was used to determine levels of BDNF in both pre-treated and after CPAP treatment OSAS patients. Flat-bottom 96 well plates were coated with the capture antibody (anti-BDNF monoclonal antibody) by incubation in 25 mM carbonate buffer, pH 9.7, overnight at + 4˚C. The plates were washed with TBST and incubated for 2 hours with 100 μl of the appropriate detection antibody (anti-BDNF polyclonal antibody). A standard curve for BDNF (7.8–500 pg/ml) was made. The optical density was measured at 450 nm using a plate reader (Bio-rad 550 Microplate Reader, Bio-Rad Laboratories, Hercules, CA, USA). All determinations were done in triplicates.

## Statistics

The student's t test was performed for unpaired and paired data depending on the case, to compare averages of two groups. Fisher exact test was used in the analysis of contingency. The

ANOVA test was used for the comparison between 3 groups. Correlation analysis were performed by Pearson coefficient "r". A statistical significance p <0.05 was defined. GraphPad Prism software version 8.2.0 (San Diego California USA) was used for statistical analysis.

## Results

According to inclusion and exclusion criteria, 24 naïve severe OSAS patients and 10 healthy control volunteers were enrolled. Subjects with OSAS had a median age of 61.9 ± 10.5 years, a mean body mass index (BMI) of 32.9 ± 6.9, a mean abdomen circumference of 118.2 ± 15.6 cm, and an average neck circumference of 44.4 ± 3.5 cm. Controls subjects had a mean age of 66.7 ± 6.6 years, a mean BMI of 32.1 ± 5.7, a mean abdomen circumference of 112.9 ± 8.6 cm and an average neck circumference of 44.1 ± 2.6 cm. The arterial blood gas analysis, the respiratory function tests and the comorbidities are collected in both groups. The baseline characteristics of the population is summarized in Table 1.

No significant difference, between demographic, anthropometric, arterial blood gas analysis, functional respiratory tests and comorbidities characteristics, in the two groups, were detected.

The severe OSAS patients had an apnea hypopnea index (AHI) and oxyhemoglobin desaturation index (ODI) values of 53.6 ± 12.3 and 55.9 ± 13.8 respectively. Control group had an ODI value was 3 ± 1.2.

The mean score of the MoCA questionnaire was 26.0 ± 1.6 for the T0 OSAS group and 28.7 ± 0.9 for the controls showing a statistically significant difference (p<0.0001) (Fig 1).

**Table 1. Demographic and clinical characteristics of the study population.**

|  | OSAS (n = 24) | CONTROL (n = 10) | p value |
|---|---|---|---|
| Age (years) [a] | 61.9 ± 10.5 | 66.7 ± 6.6 | p = 0.20 |
| Female sex, n (%) [b] | 4 (16.6%) | 2 (20%) | p>0.99 |
| BMI (Kg/m$^2$) [a] | 32.9 ± 6.9 | 32.1 ± 5.7 | p = 0.75 |
| Current smoker, n (%) [b] | 4 (16.6%) | 2 (20%) | p>0.99 |
| Neck Circumference (cm) [a] | 44.4 ± 3.5 | 44.1 ± 2.6 | p = 0.80 |
| Waist Circumference (cm) [a] | 118.2 ± 15.6 | 112.9 ± 8.6 | p = 0.33 |
| pH [a] | 7.42 ± 0.03 | 7.41 ± 0.01 | p = 0.52 |
| pCO2 (mmHg) [a] | 40.6 ± 3.17 | 39.5 ± 3.04 | p = 0.36 |
| pO2 (mmHg) [a] | 80.6 ± 7.13 | 78.7 ± 4.60 | p = 0.44 |
| FEV1/FVC (%) [a] | 80.9 ± 4.47 | 79.7 ± 3.62 | p = 0.50 |
| FEV1 (l) [a] | 3.72 ± 0.45 | 3.83 ± 0.52 | p = 0.55 |
| FVC (l) [a] | 4.28 ± 0.62 | 4.42 ± 0.70 | p = 0.57 |
| TLC (l) [a] | 6.24 ± 0.39 | 6.33 ± 0.34 | p = 0.52 |
| RV (l) [a] | 2.09 ± 0.29 | 2.12 ± 0.26 | p = 0.73 |
| Arterial hypertension, n (%) [b] | 14 (58.3%) | 6 (60%) | p>0.99 |
| Coronary Artery Disease, n (%) [b] | 4 (16.6%) | 1 (10%) | p>0.99 |
| Atrial fibrillation, n (%) [b] | 3 (12.5%) | 1 (10%) | p>0.99 |
| Dyslipidemia, n (%) [b] | 9 (37.5%) | 3 (30%) | p>0.99 |
| Hypothyroidism, n (%) [b] | 2 (8.3%) | 0 (0%) | p>0.99 |
| Diabetes mellitus II, n (%) [b] | 4 (16.6%) | 2 (20%) | p>0.99 |

BMI: Body Mass Index; FEV1: Forced Expiratory Volume in the 1 second; FVC: Forced Vital Capacity; TLC: Total Lung Capacity. RV: Residual Volume.

[a]T test. Values are presented as mean ± standard deviation.

[b]Fisher's exact test. Values are presented as number and % of the relative column.

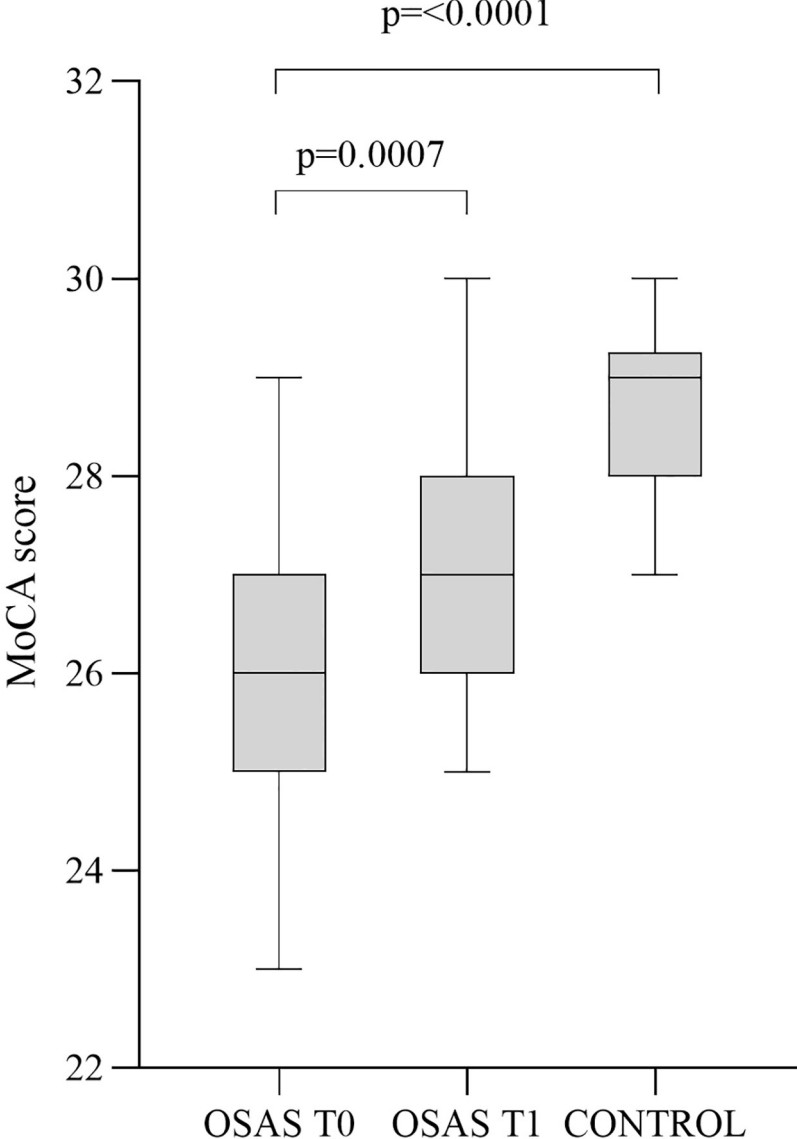

**Fig 1. Population's MoCA score.** The figure shows a whisker plot with mean and interquartile range for the different group (before and after treatment and control) which indicates a statistically significant difference between control and T0 OSAS group and an increased MoCA score after CPAP treatment in OSAS patients.

At the baseline, untreated OSAS patients (group T0) have statistically significant higher BDNF serum levels (52.2 ± 23.0 ng/ml) compared with healthy controls (29.0 ± 8.9 ng/ml) (p = 0.0053, Fig 2).

OSAS patients underwent CPAP and demonstrated a total adherence, proved by checking the ventilator memory cards. CPAP treatment lasted 90 days. CPAP treated OSAS patients (T1 group) showed a serum BDNF value of 48.5 ± 21.8 ng/ml that indicated a downward trend although not statistically significant (p = 0.13, Fig 2). The analysis of cognitive score (MoCA score) trajectory over time revealed statistically significant differences between T0 and T1 groups (p = 0.0007, Fig 1).

The Table 2 show the serum levels of BDNF and the MoCA score at each time point.

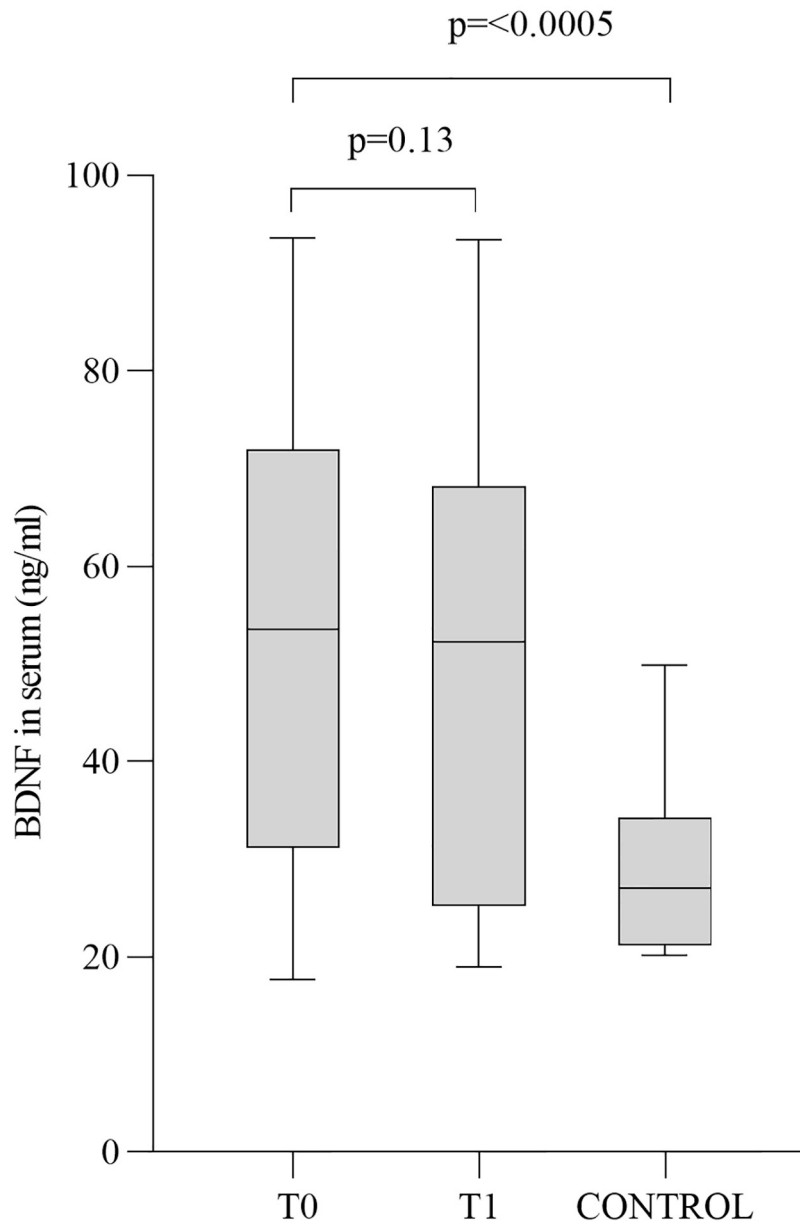

**Fig 2. BDNF serum levels at each time point.** The figure shows a whisker plot with mean and interquartile range for the different group (before and after treatment and control) which indicates a statistically significant difference between control and T0 OSAS group.

**Table 2. BDNF serum levels and MoCA score at each time point.**

|  | OSAS T0 | OSAS T1 | CONTROL |
|---|---|---|---|
| BDNF in serum (ng/ml) | 52.2 ± 23.0 | 48.5 ± 21.8 | 29.0 ± 8.9 |
| MoCA score | 26.0 ± 1.6 | 27.2 ± 1.2 | 28.7 ± 0.9 |

Data are presented as mean ± standard deviation. BDNF: Brain Derived Neutrophic Factor; MoCA: Montreal Cognitive assessment.

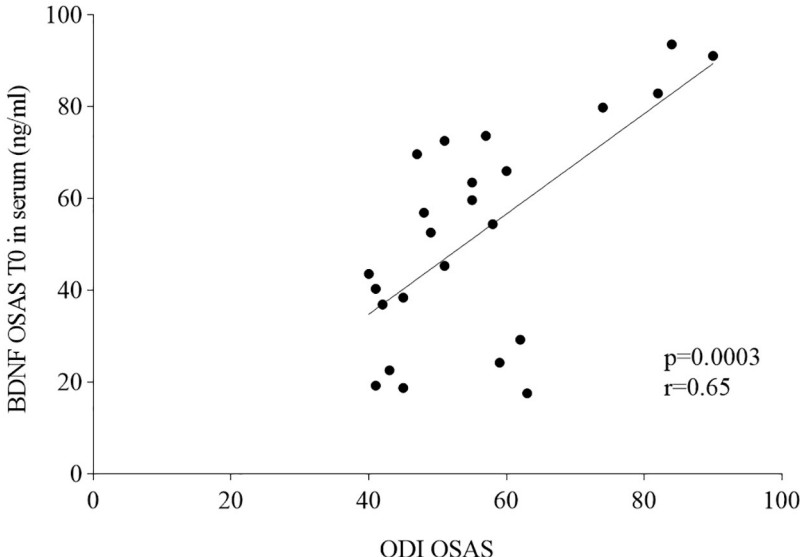

**Fig 3. Correlation between ODI and BDNF serum levels in T0 OSAS patients (before treatment).**

ODI values in OSAS patients showed a direct correlation to BDNF serum levels in T0 group (p = 0.0003; r = 0.65; Fig 3).

A positive correlation was documented comparing BDNF serum levels and the MoCA score in T0 OSAS patients by the Pearson's test (p = 0.0001; r = 0.67; Fig 4), an increase of MoCA score was always associated with more elevated BDNF serum levels.

No statistically significant correlation was demonstrated in T0 OSAS patient among BDNF and anthropometric parameters.

## Discussion

BDNF is known to play a critical role in neuroprotection and neuronal plasticity in both central and peripheral nervous system [19]. CNS endothelial cells have been shown to synthesize

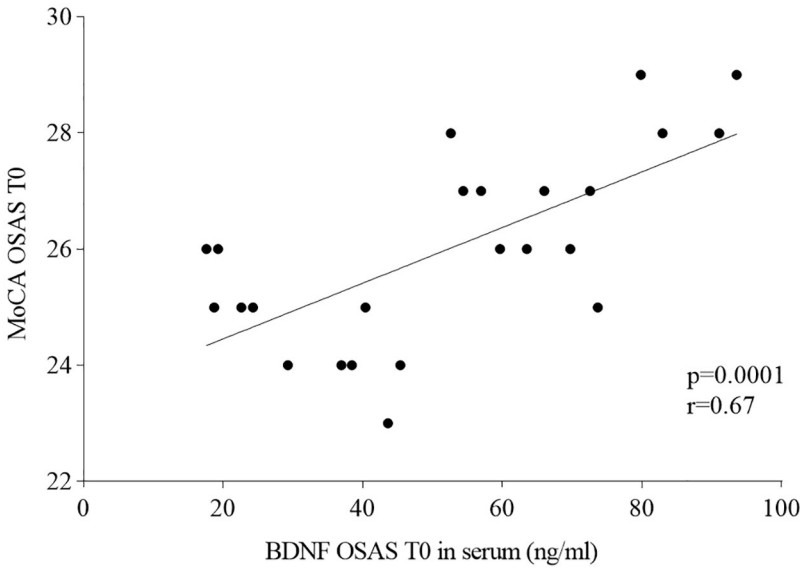

**Fig 4. Correlation between BDNF serum levels and MoCA score in T0 OSAS patients (before treatment).**

*de novo* BDNF following repeated hypoxic stimulation and patient with acute ischemic stroke have increased BDNF circulating levels [21, 22]. However, whether BDNF levels are elevated in OSAS, a pathological condition characterized by intermittent hypoxia, is still controversial. Using a rat model of repetitive acute intermittent hypoxia (rAIH) Satriotomo et al. [23] demonstrated an increased CNS synthesis of BDNF, which was important to elicit motor plasticity in respiratory and non-respiratory motor neurons.

Consistently with these findings in the animal model, here we demonstrated that patients affected with OSAS have higher circulating levels of BDNF compared to healthy control volunteers. Moreover, we also showed a positive correlation between the levels of BDNF and oxygen desaturation index (ODI) indicating that patients with more pronounced intermittent hypoxia are more likely to have higher level of circulating BDNF.

Differently from our findings, previous studies from Panaree et al. [24], Y Wang et al. [25] and Staats et. al [26] showed no differences in serum and plasma BDNF levels in OSAS patients compared to healthy control volunteers. These studies grouped together OSAS with different severity and the oxygen desaturation index (ODI) is not discussed, which can account for the different results observed. In addition, Staats et al.[26] found that the BDNF levels were dramatically reduced when a single night of CPAP was used as a treatment in these patients, indicating a possible relation between the levels of peripheral BDNF and the intermittent hypoxia in OSAS, probably caused also by an increased brain uptake. Even if discrepancies observed among the studies may be due to different clinical variables including OSAS severity and comorbidities, altogether these findings corroborate the hypothesis that the intermittent hypoxia, typical of OSAS patients, is associated with altered levels of BDNF.

OSAS is a disease characterized by rAIH and sleep fragmentation which have been demonstrated to be associated with neuro-inflammation and impaired neuro-cognitive functions [20]. Our data corroborate this association since we showed that the OSAS patients T0 had a significantly lower average MoCA questionnaire score than healthy control volunteers. However, the positive correlation that we found between the BDNF serum levels and the ODI in OSAS T0 subjects is an important evidence suggesting a reaction to the state of nocturnal rAHI. This reaction appears to be neuroprotective since we also showed a significantly positive correlation between BDNF levels and MoCA score. In particular, subjects with higher BDNF values had less impairment in neurocognitive functions. Our results are in line with previous findings from Wang WH et al. [27] who found a significant and positive correlation between MoCA score and BDNF, stating the determinant role of BDNF in neuro-cognitive protection.

Since BDNF increase appears to be an adaptive response to intermittent hypoxia, an effective CPAP treatment should be able to reduce BDNF release as observed by previous studies from Staats et al. [26] and Y Wang et al. [25] Although we have observed a mild reduction in BDNF levels in OSAS patients after CPAP treatment, this data did not reach statistical significance. This could be related to a sample size effect, to the duration of CPAP treatment and/or to a more complex and dynamic regulation of BDNF release in response to CPAP treatment.

Several evidence strongly suggest that altered BDNF signaling may be related to the pathogenesis of several neurological disorders including Huntington's disease, Alzheimer's disease, and depression [28]. Altogether our data suggest that increased levels of BDNF in OSAS patients may be an important neuroprotective reaction to the nocturnal intermittent hypoxia and sleep fragmentation aimed to limit the progressive neuro-cognitive impairment typical of this condition.

## Conclusions

Our study shows that increased levels of BDNF in OSAS patients are associated with better neurocognitive function at the MoCA test. This data shed new light in the complex

pathophysiological mechanisms of adaptation to intermittent hypoxia associated with OSAS and suggest that BDNF, released in response to intermittent hypoxia, appear to have a protective role by limiting neurocognitive impairment.

## Supporting information

**S1 Dataset. Minimal significant dataset.**
(XLSX)

## Acknowledgments

The authors thank all patients for the participation in the study.

## Author Contributions

**Conceptualization:** Krisstopher Richard Flores, Alberto Ricci.

**Data curation:** Fausta Viccaro.

**Formal analysis:** Krisstopher Richard Flores.

**Investigation:** Krisstopher Richard Flores, Fausta Viccaro, Stefania Scarpino, Arianna Di Napoli.

**Methodology:** Krisstopher Richard Flores, Alberto Ricci.

**Resources:** Mauro Aquilini, Stefania Scarpino, Francesco Ronchetti, Rita Mancini, Arianna Di Napoli.

**Supervision:** Alberto Ricci.

**Visualization:** Krisstopher Richard Flores, Fausta Viccaro.

**Writing – original draft:** Krisstopher Richard Flores, Fausta Viccaro, Francesco Ronchetti.

**Writing – review & editing:** Krisstopher Richard Flores, Fausta Viccaro, Davide Scozzi, Alberto Ricci.

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
