## [Decision Letter · Decision Letter 0]

23 Oct 2019

PONE-D-19-26889

Protective role of brain derived neurotrophic factor (BDNF) in obstructive sleep apnea syndrome (OSAS) patients

PLOS ONE

Dear Dr. Ricci,

Thank you for submitting your manuscript to PLOS ONE. After careful consideration, we feel that it has merit but does not fully meet PLOS ONE’s publication criteria as it currently stands. Therefore, we invite you to submit a revised version of the manuscript that addresses the points raised during the review process.

We would appreciate receiving your revised manuscript by Dec 07 2019 11:59PM. To enhance the reproducibility of your results, we recommend that if applicable you deposit your laboratory protocols in protocols.io, where a protocol can be assigned its own identifier (DOI) such that it can be cited independently in the future. For instructions see: http://journals.plos.org/plosone/s/submission-guidelines#loc-laboratory-protocols

We look forward to receiving your revised manuscript.

Kind regards,

Claudio Tripodo

Academic Editor

PLOS ONE

Journal Requirements:

1.

Additional Editor Comments (if provided):

Dear Dr Ricci, your manuscript has been reviewed by one expert Referee and by a member of the Editorial Board of the Journal. Both Reviewers found you manuscript of interest. I would recommend that in your revision you provide the requested additional detailes about the patients' cohort. Please also consider a thorough editing of the language and check the display items for consistency.

Reviewers' comments:

Reviewer's Responses to Questions

**Comments to the Author**

1. Is the manuscript technically sound, and do the data support the conclusions?

Reviewer #1: Yes

2. Has the statistical analysis been performed appropriately and rigorously? 

Reviewer #1: Yes

3. Have the authors made all data underlying the findings in their manuscript fully available?

Reviewer #1: Yes

4. Is the manuscript presented in an intelligible fashion and written in standard English?

Reviewer #1: Yes

5. Review Comments to the Author

Reviewer #1: Dr. Alberto Ricci and coworkers designed a study aimed at evaluating BDNF in patients with OSAS, compared with healthy controls.

To better characterize the two groups and to verify possible factors influencing BDNF, the Authors should present more details on the two populations including respiratory function data (spirometry) and presence of comorbidities.

6. PLOS authors have the option to publish the peer review history of their article (what does this mean?). If published, this will include your full peer review and any attached files.

Reviewer #1: No

---

## [Author Response · Author response to Decision Letter 0]

12 Dec 2019

We have greatly appreciated the referees’ comments and suggestions. Their constructive criticisms, allowed us to improve the quality and the significance of the results of the submitted manuscript. 

Our revision was detailed below.

The manuscript respects the PLOS ONE’s style requirements.

A carefully review of the manuscript by an English mother tongue has been done. Actually, the manuscript is technically sound and the data does support the conclusions.

The statistical analysis was appropriately and rigorously reviewed.

We make all data underlying the findings fully available.

We added additional patients’ cohort details including respiratory function findings, blood gas analysis and presence of comorbidities. Therefore, a new Table 1 was added to the study.

---

## [Editor Report · Decision Letter 1]

31 Dec 2019

Protective role of brain derived neurotrophic factor (BDNF) in obstructive sleep apnea syndrome (OSAS) patients

PONE-D-19-26889R1

Dear Dr. Ricci,

We are pleased to inform you that your manuscript has been judged scientifically suitable for publication and will be formally accepted for publication once it complies with all outstanding technical requirements.

With kind regards,

Claudio Tripodo

Academic Editor

PLOS ONE

Additional Editor Comments (optional):

Please carefully review all the data and iconographic content of your manuscript and check legends for consistency.
---

## [Editor Report · Acceptance letter]

10 Jan 2020

PONE-D-19-26889R1 

Protective role of brain derived neurotrophic factor (BDNF) in obstructive sleep apnea syndrome (OSAS) patients 

Dear Dr. Ricci:

I am pleased to inform you that your manuscript has been deemed suitable for publication in PLOS ONE. Congratulations! Your manuscript is now with our production department. 

With kind regards,

on behalf of

Dr. Claudio Tripodo 

Academic Editor

PLOS ONE